

# Evolutionary and functional implications of hypervariable loci within the skin virome

Geoffrey D. Hannigan[1], Qi Zheng[1], Jacquelyn S. Meisel[1], Samuel S. Minot[2], Frederick D. Bushman[3] and Elizabeth A. Grice[1,3]

[1] Department of Dermatology, University of Pennsylvania, Philadelphia, PA, USA
[2] One Codex, San Francisco, CA, USA
[3] Department of Microbiology, University of Pennsylvania, Philadelphia, PA, USA

## ABSTRACT

Localized genomic variability is crucial for the ongoing conflicts between infectious microbes and their hosts. An understanding of evolutionary and adaptive patterns associated with genomic variability will help guide development of vaccines and antimicrobial agents. While most analyses of the human microbiome have focused on taxonomic classification and gene annotation, we investigated genomic variation of skin-associated viral communities. We evaluated patterns of viral genomic variation across 16 healthy human volunteers. Human papillomavirus (HPV) and *Staphylococcus* phages contained 106 and 465 regions of diversification, or hypervariable loci, respectively. *Propionibacterium* phage genomes were minimally divergent and contained no hypervariable loci. Genes containing hypervariable loci were involved in functions including host tropism and immune evasion. HPV and *Staphylococcus* phage hypervariable loci were associated with purifying selection. Amino acid substitution patterns were virus dependent, as were predictions of their phenotypic effects. We identified diversity generating retroelements as one likely mechanism driving hypervariability. We validated these findings in an independently collected skin metagenomic sequence dataset, suggesting that these features of skin virome genomic variability are widespread. Our results highlight the genomic variation landscape of the skin virome and provide a foundation for better understanding community viral evolution and the functional implications of genomic diversification of skin viruses.

Corresponding author
Elizabeth A. Grice,
egrice@upenn.edu

## INTRODUCTION

Localized genomic modifications are ammunition in the ongoing battle between hosts and infectious agents. The human adaptive immune response relies on localized genomic diversity of antigen receptors to facilitate detection and efficient removal of foreign agents (*Borghans, Beltman & De Boer, 2004*; *Kubinak et al., 2012*). Infectious microbes, such as bacteria and viruses, likewise rely on genomic variation to modulate tropism and facilitate immune evasion (*Malim & Emerman, 2001*; *Doulatov et al., 2004*; *Minot et al., 2012*; *Schillinger et al., 2012*; *Das et al., 2013*; *Minot et al., 2013*; *Guo et al., 2014*).

Potential selective benefits of targeted variation in viruses include immune evasion and widening of host tropism (*Borghans, Beltman & De Boer, 2004*; *Kubinak et al., 2012*).

Most contemporary low resolution studies of the human microbiome evaluate functional potential through taxonomic classification and whole gene identification (*Schloss & Handelsman, 2008*; *Human Microbiome Project Consortium, 2012*; *Langille et al., 2013*; *Hannigan et al., 2014*; *Ly et al., 2014*; *Norman et al., 2015*; *Lim et al., 2015*; *Meisel et al., 2016*). These approaches are usually unable to capture nucleotide variations that affect functionality of proteins encoded in the microbiome, which can be altered by differences in only a few nucleotides. For example, viruses such as *Bordetella* bacteriophages, hepatitis C virus, and others only require short variable regions within a gene to facilitate functional changes in processes including tropism diversity, immune evasion, drug resistance, and adaptation to host auxotrophies (*Bacher, Bull & Ellington, 2003*; *Doulatov et al., 2004*; *Donaldson et al., 2010*; *Guan et al., 2012*; *Shah et al., 2014*). Contemporary low-resolution studies also fail to identify genetic cassettes that promote targeted diversity, such as diversity generating retroelements (DGRs). DGRs promote targeted genetic diversification in bacteriophages through error-prone cycles of transcription, reverse transcription, and integration; through this process information encoded in a non-variable template region is copied in a fallible fashion into a variable region within a coding sequence (*Doulatov et al., 2004*; *Minot et al., 2012*; *Schillinger et al., 2012*).

Here, we investigate skin virome evolution and adaptation by inferring the selective pressure, functional diversity, and substitution patterns associated with targeted hypervariation. We focus on three prominent cutaneous viruses: human papillomavirus (HPV), *Propionibacterium* phage, and *Staphylococcus* phage. HPV is associated with the development of skin cancer, especially in immune-suppressed individuals (*Vinzón et al., 2014*; *Wang et al., 2014*; *Quint et al., 2015*). Current vaccine efforts aim to target conserved antigens for broad strain protection—thus a greater understanding of HPV genomic diversity could improve design of vaccines (*Schiller & Lowy, 2012*; *Vinzón et al., 2014*). *Staphylococcus* phages can modulate *Staphylococcus* pathogenic gene expression and facilitate transmission of antibiotic resistance (*Bae et al., 2006*; *Varga et al., 2012*). *Propionibacterium* phages are associated with *Propionibacterium acnes* and have therapeutic potential for treating acne (*Marinelli et al., 2012*; *Hannigan & Grice, 2013*). Our findings build upon previous analyses of individual virus genomic variability and provide new insight into phage biology of the cutaneous microbiome.

## MATERIALS AND METHODS

### Analysis details and availability

All associated source code and explanatory README files are available for review at the following GitHub repository: https://github.com/Microbiology/ViromeVarScripts.

### Data acquisition and quality control

The primary skin virome dataset was acquired from SRA accession: SRP049645 (*Hannigan et al., 2015*) and includes sequences from samples collected at the second and

third time points. The secondary dataset was obtained from *Oh et al. (2014)* (SRA BioProject: 46333). Retroauricular crease samples were downloaded from the NCBI SRA BioProject: 46333. For samples from both the primary and secondary dataset, sequences were trimmed with the FASTX-Toolkit (version 0.0.14), using a quality score cutoff of 33. Remaining sequences with similarity to the human genome were removed using the standalone DeconSeq toolkit (version 0.4.3) (*Schmieder & Edwards, 2011*).

## Contig assembly and taxonomic identification

Contigs from the primary dataset were obtained from the published Figshare source (DOI 10.6084/m9.figshare.1281248). Contigs from both the primary and secondary datasets were separately assembled using the Ray metagenomic assembly software, specifying a minimum contig length of 500 bp and otherwise default parameters (v2.3.1) (*Boisvert et al., 2012*). Within each dataset, sequences from all samples were combined prior to assembly to facilitate the most complete contig assembly. Contig coverage was determined by aligning sequences back to the contigs with the bowtie2 toolkit (v2.1.0; seed substring length of 25 and one mismatch allowed in alignment) (*Langmead & Salzberg, 2012*). Quantification of reads mapping back to contigs was obtained by parsing bowtie2 output using Perl and BASH scripts presented in the supplemental source code. Coverage was calculated using the number of reads mapping to each contig. The blastn program from the NCBI Blast+ toolkit (version 2.2.0) was used to determine similarity of contigs to virus reference genomes (*Camacho et al., 2009*). Contigs were blasted against a previously described virus-specific genome reference database, which is a subset of the EMBL reference genome database (*UniProt Consortium, 2014*; *Hannigan et al., 2015*). A similarity threshold of $e$-value $< 10^{-3}$ was used, and sequences with multiple potential identities were resolved by using only hits with the lowest $e$-values. Although this was the minimum threshold, the contigs of interest exhibited $e$-values $< 10^{-3}$.

## Phylogenetic analysis

We constructed phylogenies using the L1 capsid gene for HPV (*Ma et al., 2014*) and the large terminase subunit for the *Staphylococcus* and *Propionibacterium* phages (*Gutiérrez et al., 2013*; *Ma et al., 2014*) as phylogenetic marker genes. For reference, we used the PAVE reference L1 genes (https://pave.niaid.nih.gov/, accessed 2015-06-03) (*Van Doorslaer et al., 2013*). The large terminase subunit references for *Staphylococcus* and *Propionibacterium* phages were from the NCBI gene sequence database (*Staphylococcus* phage: accessed 2015-09-14, search term: *((phage terminase large subunit staphylococcus)) AND "viruses"[porgn:__txid10239] NOT "ORF" NOT "hypothetical" NOT "putative;"* *Propionibacterium* phage: accessed 2015-09-15, search term: *((phage terminase large subunit propionibacterium)) AND "viruses"[porgn:__txid10239]*). To extract the phylogenetic marker genes from the virome contigs, we determined which open reading frames (ORFs) matched the reference genes by nucleotide similarity (nucleotide blast, $e$-value 1e−10). Only ORFs longer than 1.2 kb were included in the analysis. The average reference gene lengths were all longer than this threshold (average reference gene length: HPV = 2,519 bp, *Staphylococcus* phage = 1,307 bp, *Propionibacterium* phage = 1,511 bp).

Contig and reference marker genes were aligned using the Smith–Waterman algorithm and 1,000 iterations as implemented by the mafft aligner (v7.221) (*Katoh & Standley, 2013*). Phylogeny was constructed using RAxML (version 8.1.21) (*Stamatakis, 2014*). The phylogenetic tree was visualized using Figtree (*Rambaut, 2006*).

## Identification of temperate phage contigs

As has been described previously, we identified temperate (lysogenic) phage contigs using three genomic markers: contig nucleotide similarity to (1) phage integrase genes, (2) prophage genes within the ACLAME prophage database, and (3) bacterial reference genomes. We performed a blastx alignment (*e*-value 1e−10, percent identity threshold 75%) of the genes within the ACLAME prophage database (*Leplae, Lima-Mendez & Toussaint, 2010*), a blastx alignment with integrase genes from Uniprot database, and a blastn alignment of the *Staphylococcus* phage contigs to *Staphylococcus* bacterial reference genomes. Integrase genes were obtained from the online Uniprot database using the search term "*organism:phage gene:int NOT putative.*" *Staphylococcus* reference genomes were obtained from the NCBI nucleotide database using the search term "'Staphylococcus'[Organism] AND 'complete genome'[Name] NOT phage[All Fields] NOT contig[All Fields] NOT ('unidentified plasmid'[Organism] OR plasmid[All Fields]) AND (bacteria[filter] AND biomol_genomic[PROP])." Both were accessed December 22, 2016. Together this allowed us to detect regions of contigs that demonstrated a high similarity to temperate phage gene signatures.

## Identification of hypervariable loci

The bowtie2 alignments of reads to viral contigs were formatted (e.g., conversion from binary to ASCII format) and then single nucleotide polymorphisms (SNPs) were called using VarScan (v2.3.7) (*Li et al., 2009*; *Koboldt et al., 2012*). The "pileup2snp" program from VarScan was used with a minimum minor allele frequency threshold of 1%, a read depth of 8, and a minimum of two supporting reads for variant calls. Indels were excluded.

To identify hypervariable loci, we used a geometric distribution based statistic approach as described previously (*Zheng et al., 2010*), which, compared to sliding window searches and other similar methods, has the advantage of avoiding boundary difficulties and variations within contigs. We used a geometric distribution to model the probability of achieving two SNPs separated by a specified non-SNP nucleotide distance. Each between-SNP distance was associated with a probability and the probability of a particular distance occurring by randomly sampling was less than 5%. Thus, we identified a range of SNP distances as significantly less than background if they occurred within our dataset less than 5% of the time.

## Protein family domain identification within hypervariable loci ORFs

Protein family domains were identified in ORFs that contained hypervariable loci. The subset of translated virus ORFs that contained hypervariable loci were aligned to the standard Pfam protein family domain database using hmmscan within the HMMer toolkit (version 3.1) and GA gathering bit score thresholds (*Finn, Clements & Eddy, 2011*).

## Prediction of single amino acid variant effect on phenotype

The SuSPect algorithm was employed to predict the likelihood of SNP-associated single amino acid variants (SAVs) impacting phenotype. We used SuSPect to create a matrix of likelihood scores for every possible SAV at every position in the ORFs that contained hypervariable loci. This matrix was used as a reference to quantify the likelihood of each hypervariable loci SNP to impact the resulting phenotype. The significance of the score differences between viruses was calculated using a Wilcoxon rank-sum test.

## Evolutionary pressure of hypervariable loci and virus genomes

We assessed the evolutionary pressure of a gene using the $pN/pS$ ratio as in Formula 1, where $M_N$ and $M_S$ represent the observed number of non-synonymous and synonymous SNPs, respectively. These values were normalized by the total number of possible non-synonymous or synonymous substitutions ($N_i$ and $S_i$, respectively), in order to avoid potential codon usage bias. Furthermore, to normalize for sequence coverage of the SNPs and prevent extreme values, a pseudocount value of an arbitrarily small number was added to the $M_N$ and $M_S$ values, which was defined as half of the square root of the median sequence coverage of SNPs within the region of interest ($C_M$). The pseudocount approach was used to prevent infinite and illegal values when $M_N$ or $M_S$ had zero values, thus allowing consideration of otherwise infinite or ignored data points. For example, an absence of synonymous mutations would result in an infinitely large value (dividing by zero) thus forcing exclusion of the data point. Our approach preserves this data and allows us to draw conclusions from the largest possible dataset, and has been shown to be effective in previous studies (*Novaes et al., 2008*; *Bajgain et al., 2011*).

$$\frac{pN}{pS} = \frac{\left( \frac{M_N + \frac{\sqrt[2]{C_M}}{2}}{\sum_{i=1}^{L} \frac{N_i}{3}} \right)}{\left( \frac{M_S + \frac{\sqrt[2]{C_M}}{2}}{\sum_{i=1}^{L} \frac{S_i}{3}} \right)} \qquad (1)$$

The formula used to calculate the $pN/pS$ ratio for a gene. $M_N$ is the number of non-synonymous SNPs within the gene and $M_S$ is the number of synonymous SNPs found within the gene. Each mutation value is normalized for the likelihood that the result would have happened by chance, calculated as the sum of the proportions of nucleotides that would have resulted in either a non-synonymous or synonymous mutation. To calculate this proportion, the possible non-synonymous mutations ($N$) and synonymous mutations ($S$) at position i within the gene are expressed as a fraction of the three possible alternate nucleotides. SNP counts were smoothed as pseudo-counts using the median SNP sequence coverage ($C_M$).

This analysis is similar to the $dN/dS$ calculations often performed to estimate degrees of natural selection among genomes (*Nishida, Frith & Nakai, 2009*; *Schloissnig et al., 2013*).

It is important to note however that such an analytical approach would be inappropriate for this type of sample set because the nucleotide variations are not assignable to isolated strains, which prevents haplotype identification that is a necessary component of *dN/dS* calculations. The *pN/pS* calculation does not assume haplotypes, and is therefore appropriate for metagenomic datasets.

To estimate the selective pressure on hypervariable loci, the locations of the hypervariable loci were extracted, along with their immediately adjacent regions, using a Perl script as presented in the supplemental source code. Adjacent regions are defined as the genomic regions that are two times the length of the hypervariable loci and located immediately before and after the hypervariable loci. These positions were used with the contig sequences, SNP call data, and *pN/pS* calculator to estimate their selective pressure. Hypervariable regions outside of coding regions were not considered.

Calculation of the overall selective pressure on virus contigs was performed in a similar approach to the hypervariable loci selective pressure. Predicted ORFs were first extracted from the contigs using the Glimmer3 toolkit (v3.02) (*Delcher et al., 2007*). The predicted ORFs, along with the contig sequences, SNP profile, and *pN/pS* calculator were used to calculate the overall selective pressure on each gene within each contig. The distributions of selective pressures observed for each gene were observed as categorized by virus type.

## Amino acid frequency, charge, and polarity

Amino acid abundance profiles were calculated while correcting for the random probability of that substitution. More specifically, each value was weighted for the number of nucleotides that result in the same amino acid as weighted value = *((number of nucleotide substitutions resulting in same amino acid)/3)$^{-1}$*. Relative abundance was calculated as the sum of the corrected frequencies. Charge and polarity were determined using a simple table of known amino acid properties. Differences in profiles between viruses were calculated using a chi-square test.

## Diversity generating retroelement identification

We identified potential DGRs by collecting assembled contigs that contained ORFs similar to known reverse transcriptase genes, and a duplicated nucleotide region less than 150 bp in length. Reverse transcriptase ORFs were identified using blastx (*e*-value < $10^{-5}$) and the Uniprot reference reverse transcriptase sequences (http://www.uniprot.org/uniprot/?sort=score&desc=&compress=yes&query=%22reverse%20transcriptase%22%20(phage%20OR%20virus)&fil=&format=fasta&force=yes). Repeat regions were identified by comparing each contig to itself with tblastx (*e*-value < $10^{-50}$) and were filtered using custom scripts to remove duplicates and regions longer than 150 bp. DGR candidates were removed if they contained no hypervariable loci or if the variable region was not within a predicted ORF.

Diversity generating retroelements were visualized in the Integrated Genomic Viewer using the DGR cassettes and bowtie2 aligned sequences described above. The linkage disequilibrium was calculated using a custom Perl script for formatting and the

"LDheatmap" and "genetics" R packages for analysis and visualization (*Shin et al., 2006*; *Warnes et al., 2003*). The linkage disequilibrium for each pair of SNPs was calculated as the squared allelic correlation ($R^2$).

## Comparison of primary analysis to validation dataset

Near identical contigs were identified between the primary and secondary validation dataset by aligning the two individually assembled contigs to each other using bowtie2, with a specified seed length of 25 and up to one seed mismatch. Sequences from our primary dataset and the Oh et al. *(2014)* dataset were aligned to the near identical contigs. These alignments were used to identify shared SNP locations between our dataset and the Oh et al. *(2014)* dataset. We quantified shared SNP location as percent of our primary analysis SNPs whose location was identical to those of SNPs in the secondary validation dataset. As a control, we compared these results to a simulated dataset where SNP position was randomly assigned. The example SNP alignment over the circular contig was generated using Genious (*Kearse et al., 2012*).

## RESULTS

### Diversity of skin viruses

We evaluated genomic variability associated with dsDNA skin viruses using a previously published human skin virome metagenomic dataset, consisting of 260,714,906 high quality sequences assembled into >76,000 contigs from 16 individuals (SRA accession: SRP049645) (*Hannigan et al., 2015*). We relied on database virus annotation to identify the taxonomic groups whose contigs had the overall highest confidence matches to reference genomes. Because greater sequencing coverage allows for more refined detection of variable nucleotides (*Schloissnig et al., 2013*), we focused our analysis on taxa whose de novo assembled contigs had sufficient coverage (greater than $10\times$). Contigs meeting these criteria were identified as *Propionibacterium* phages (contig count = 45), *Staphylococcus* phages (contig count = 319), and human papillomaviruses (HPVs; contig count = 56; Fig. 1A), representing a total of 420 contigs to be used out of the more than 76,000 total contigs in the study (Fig. S1). All of the contigs from these three taxa were used in our analysis, including those below our coverage threshold, since contigs can have regions of high coverage despite an average low coverage. More specific, secondary filtering was done while identifying SNPs. Some contigs were identified as *Pseudomonas* phages or *Enterobacteria* phages, but these taxa excluded from the analysis because their annotations were lower confidence and contig representation was minimal (Fig. 1A). The uneven virus population coverage is potentially a reflection of our inability to taxonomically identify the majority of virome sequences, and as a result we lose this information to the viral "dark matter" (*Hannigan et al., 2015*).

We evaluated the diversity of the skin viruses by constructing phylogenic trees based on conserved viral genes. Only fully assembled ORFs were considered of >1.5 kb for HPV, and >0.15 kb for *Staphylococcus* and *Propionibacterium* phages. Similar to previous studies, we used the L1 major capsid gene to classify HPV strains (*de Villiers et al., 2004*; *Ma et al., 2014*). The terminase large subunit gene was extracted from *Staphylococcus*

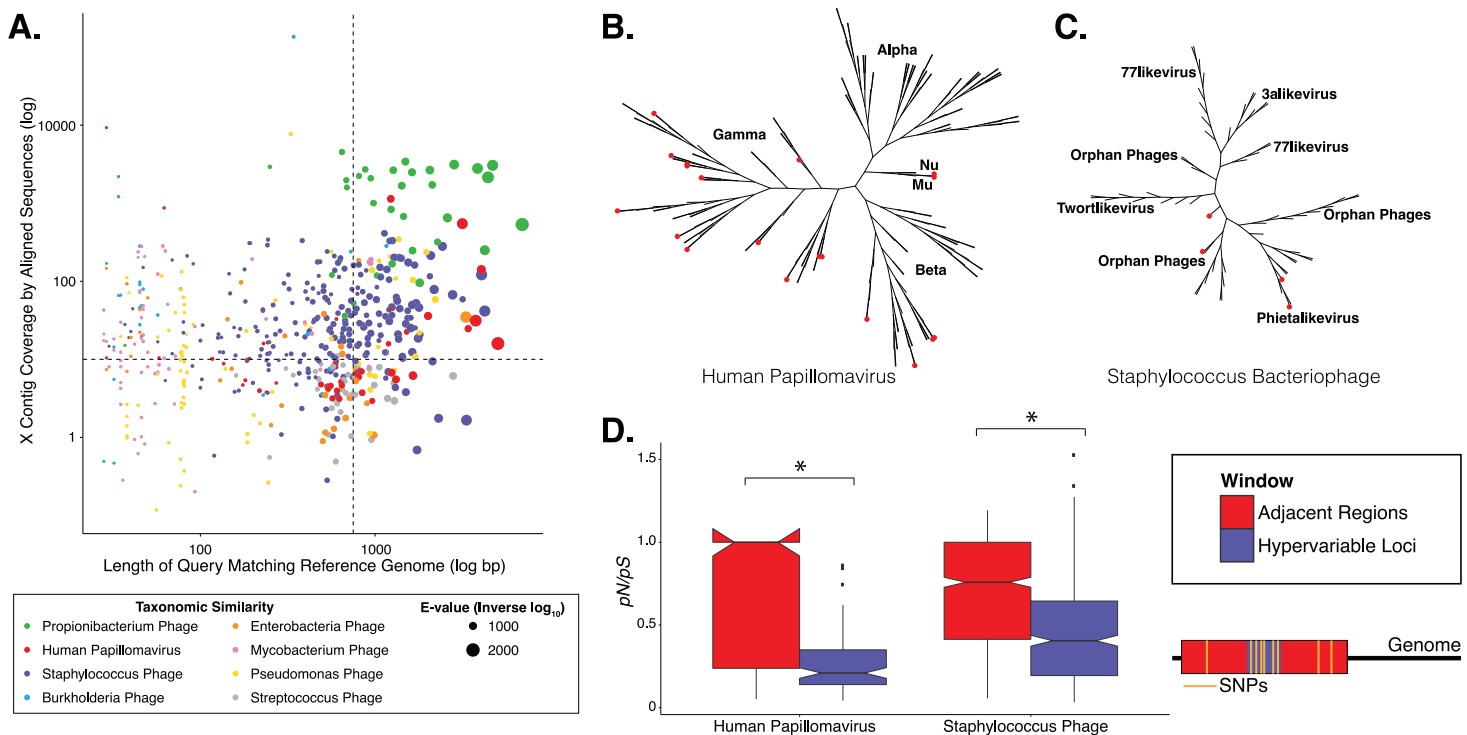

**Figure 1 Phylogenetic & evolutionary characteristics of skin virome hypervariable loci.** (A) Scatter plot depicting the candidate contigs considered for analysis in this study. Each point is a contig that mapped to a reference virus genome. The *x*-axis shows the length (log$_{10}$ scale) of the contig subsection that mapped to the reference genome. The *y*-axis shows the overall coverage of the contig, as a quantification of sequences aligning to the contig. The color highlights the reference virus genome that the contig was most similar to, and the size depicts the *e*-value (inverse log$_{10}$) associated with the contig-reference match. The horizontal dashed line marks the threshold of 10× coverage, and the vertical dashed line marks the 750 bp length threshold. (B) Phylogenetic tree of skin virome HPVs and (C) *Staphylococcus* phages, structured onto a standard phylogenetic tree using reference genomes. HPV phylogeny was based on the L1 major capsid gene and *Staphylococcus* phage phylogeny was based on the large terminase subunit. Contigs from this study are highlighted as orange dots, and genera are labeled with text. Phylogenetic lengths were normalized to ranks to facilitate visualization. (D) Box plots depicting the evolutionary pressure of HPVs (left) and *Staphylococcus* bacteriophages (right) at the hypervariable loci (blue) and the regions immediately adjacent to the hypervariable loci (red). Adjacent regions were calculated as being twice the length of the hypervariable loci (see visualization to the right). The hypervariable locus and adjacent region (combination of both sides) from each sample were evaluated for evolutionary pressure (*y*-axis) using SNPs (pink lines in right illustration). Asterisk indicates a statistically significant difference ($p < 0.01$). Notched boxplots were created using ggplot and show the median (center line), the inter-quartile range (IQR; upper and lower boxes), the highest and lowest value within 1.5 × IQR (whiskers), and the notch which provides an approximate 95% confidence interval as defined by 1.58 × IQR/sqrt($n$).

phage contigs to construct phylogeny as described previously (*Gutiérrez et al., 2013*; *Ma et al., 2014*). Because this gene is used for phylogeny of a variety of phages, we attempted to construct *Propionibacterium* phage phylogeny in a similar manner (*Ganz et al., 2014*; *Li et al., 2014*), but were ultimately unsuccessful due to the lack of a full-length de novo assembled reference genes in the dataset.

Most skin HPVs were identified as gamma HPVs, the prototypical cutaneous HPV class (Fig. 1B) (*Mistry, Wibom & Evander, 2008*). Few contigs were identified as beta and Mu/Nu HPVs, and none were identified as alpha HPVs. This is consistent with data from the Human Microbiome Project cohort (*Ma et al., 2014*).

Fewer *Staphylococcus* phage marker genes were identified, compared to HPVs, likely because *Staphylococcus* phage genomes are orders of magnitude longer than HPV

genomes, thereby decreasing the probability that contigs covered the entire genome. Because multiple displacement amplification (MDA) was not used to create this dataset, there is no MDA-associated bias toward small circular genomes. The *Staphylococcus* phage contigs belonged to the Phietalikevirus genus and orphan virus groups (those that have not yet been classified) (Fig. 1C). Of the *Staphylococcus* phage contigs identified, 49.6% (123 out of 248 contigs) were predicted to be lysogenic, based on similarity to lysogenic phages in the ACLAME database, integrase genes in the Uniprot database, and Staphylococcus reference genomes from the NCBI nucleotide database, as described previously (*Leplae, Lima-Mendez & Toussaint, 2010*; *Minot et al., 2011*; *Hannigan et al., 2015*). This is a minimum estimate of contig lysogeny, as some of the other contigs may have lysogenic signatures that we failed to identify. Furthermore, because this classification strategy is based on blast assignments, it may result in false positives if genes in the database are homologous to genes present in lytic phages.

## Hypervariable loci within the skin virome

We implemented a geometric distribution-based approach to identify regions of high genomic diversity, as in (*Zheng et al., 2010*). Regions within each contig that contained a significantly higher frequency of SNPs over the stochastic background were identified as viral hypervariable loci. Significance was defined as the frequency of SNPs having less than a 5% chance of randomly occurring, given the geometric distribution of the dataset. HPVs and *Staphylococcus* phages maintained 106 and 465 hypervariable loci, respectively. We were unable to detect hypervariable loci in the *Propionibacterium* phage population.

To determine the virus protein family domains hosting hypervariable loci, we used the hidden Markov model analysis implemented by HMMer (*Finn, Clements & Eddy, 2011*). Hypervariable loci-containing HPV genes include E6, E2, and E1 genes, which are associated with infectious gene expression, and the L1 major capsid gene, which is involved in tropism and host immune evasion (Table S1). The L1 major capsid protein is also a target in contemporary, widely used HPV vaccines (*Schiller & Lowy, 2012*). Hypervariable loci were detected in a variety of *Staphylococcus* phage genes with predicted functions related to tropism, host immune evasion, and utilization of host resources (Table S2).

## Selective pressures on hypervariable loci

We evaluated the selective pressures on virus genes by calculating the *pN/pS* ratio of non-synonymous SNPs (*pN*) to synonymous SNPs (*pS*) within each virus taxa (*Schloissnig et al., 2013*). This was used as an alternative to *dN/dS* because *dN/dS* assumes haplotype information which cannot be fulfilled by metagenomic data (*Schloissnig et al., 2013*). In the *pN/pS* calculation, neutral evolution is defined as an equal frequency of synonymous and non-synonymous polymorphisms. Selective pressure favors non-synonymous mutations, resulting in increased *pN/pS* ratios. Purifying selection has the opposite effect. Because the existing model (*Schloissnig et al., 2013*) is susceptible to stochastic effects and extreme outliers (e.g., division by zero when *pS* = 0), we added a pseudocount correction (Formula 1).

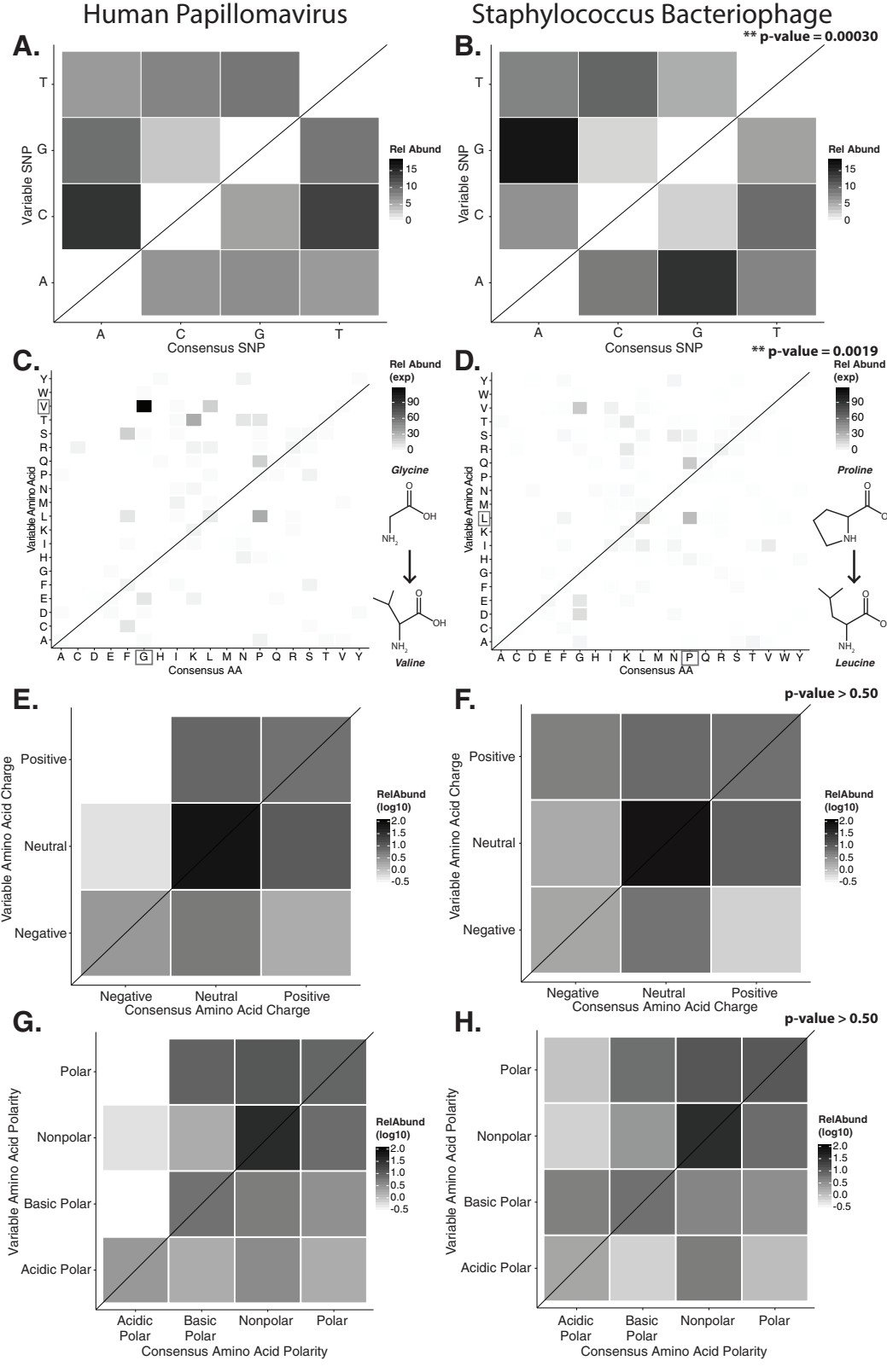

**Figure 2 Nucleotide and amino acid substitution patterns within viral hypervariable loci.** Heat maps portraying the counts of every possible nucleotide substitution for each SNP found within (A) HPV and (B) *Staphylococcus* phage hypervariable loci. Tile color weight corresponds to the relative abundance of SNP substitution counts. The diagonal line highlights the panels associated with no substitution. The substitution patterns of amino acids at each SNP are also shown with exponential transformation (C, D). An illustration of the major amino acid substitutions are provided beneath the legends as a reference. Amino acid charge (E, F) and polarity with acidity (G, H) are shown with $\log_{10}$ transformation. The absence of a basic or acidic polar identifier indicates the amino acid 20 is polar but neutral. The HPV substitution profiles are found in the left column and the *Staphylococcus* phage profiles are found on the right. Chi-square significance *p*-value, comparing variation profiles between the viruses in each row (i.e., A and B), is shown in the upper right corner of the associated *Staphylococcus* phage variation profile. The most frequently substituted amino acid pairs are highlighted with a box around the amino acid letters.

We determined whether hypervariable loci are in fact loci of focused selective pressure by comparing *pN/pS* values of the loci to the adjacent genomic regions. *pN/pS* values of hypervariable loci were significantly lower than adjacent regions in both HPV (median: adjacent = 1.0, hypervariable loci = 0.21; *p*-value = 3.4e−17) and *Staphylococcus* phage (median: adjacent = 0.76, hypervariable loci = 0.41; *p*-value = 1.8e−40) genomes, suggesting purifying selection and a propensity to maintain existing protein sequences (Figs. 1D). HPV hypervariable loci were under significantly greater purifying selection than those of *Staphylococcus* phages (median: HPV = 0.21, *Staphylococcus* phage = 0.41; *p*-value = 4.64e−9) (Fig. S1). Furthermore, not only are the *pN/pS* values of the hypervariable significantly lower than their adjacent regions, but very few of the loci have a *pN/pS* value greater than one.

To evaluate whether the observed selective pressure in HPV and *Staphylococcus* virus communities is genome-wide or localized to hypervariable loci, we quantified the selective pressure on each virus' genome by calculating the overall *pN/pS* ratio including hypervariable loci and non-hypervariable loci SNPs. We observed nearly neutral pressure across HPVs and *Staphylococcus* phages that mirrored pressures to those observed in the regions adjacent to the hypervariable loci (median: HPV = 1.0, *Staphylococcus* phage = 0.81, *p*-value = 3.2e−5) (Fig. S2).

## Functional implications of targeted substitutions within hypervariable loci

In order to evaluate the specific nucleotide changes occurring at hypervariable loci, as well as to evaluate the implications of specific nucleotide polymorphisms, we quantified the frequency of individual nucleotide substitutions within hypervariable loci. A>C and T>C substitutions were most frequent in HPV hypervariable loci (Fig. 2A). *Staphylococcus* phages exhibited a significantly different substitution profile (*p*-value = 0.00018, chi-square test), with the most common substitutions being A>G and G>A transitions (Fig. 2B). HPV and *Staphylococcus* phage substitutions were more likely to be transitions, with a transition/transversion (ti/tv) ratio of 3.25 and 2.02, respectively.

We predicted how hypervariable loci SNPs might affect protein functionality by evaluating patterns of the amino acid substitutions while correcting for the random chance that the substitution will occur. The most frequent non-synonymous amino acid

substitution in HPVs was glycine (consensus amino acid) to valine (variant amino acid, Fig. 2C). While these amino acids are (non-polar and hydrophobic), glycine is less hydrophobic than valine. The most frequent non-synonymous amino acid substitution in *Staphylococcus* phages was proline to leucine (Fig. 2D), a substitution between a non-polar cyclic amino acid and an aliphatic straight chain amino acid. Profiles of amino acid substitution were significantly different between HPVs and *Staphylococcus* phages (*p*-value = 0.0021; chi-square test).

Amino acid polarity and charge were largely maintained in HPV hypervariable loci (Figs. 2E and 2G). In instances of altered charge, visual inspection suggests the most frequent changes were from neutral to positive or negative charge, or positive to neutral charge. Consensus acidic polar residues were not associated with polymorphisms. *Staphylococcus* phage community hypervariable loci appeared to be under weaker substitution selection, with a greater diversity in amino acid charge and polarity (Figs. 2F and 2H) compared to HPV. Patterns of substitution charge and polarity were not significant (*p*-value > 0.5; chi-square test) when comparing the entire HPV to *Staphylococcus* phage substitution profiles.

We reinforced the observed functional implications of hypervariable loci by predicting the effects of their associated SAVs on gene phenotype using the support vector machine algorithm implemented in SuSPect (*Yates et al., 2014*). This method assigns a deleterious score to each hypervariable loci SNP-associated SAV, with 0 representing a neutral SAV and 100 representing a SAV with high likelihood to impact phenotype. These scores are based on the predicted impact of the SAV on the tertiary and secondary structure of the resulting protein, the location of the SAV within the resulting protein (surface vs core), and whether the SAV has previously been associated with altered protein–protein interactions. Both *Staphylococcus* phages and HPVs have an abundance of SNPs associated with SAVs predicted to be deleterious (deleterious scores approaching 100) (Fig. 3). The HPV SNPs were predicted to be significantly more likely to impact phenotype than the *Staphylococcus* phage SNPs (median: HPV = 45, *Staphylococcus* phage = 17; *p*-value < 2.2e−16), suggesting that SNPs impact functionality differently between viruses.

## Diversity generating retroelements as a mechanism for targeted hypervariability

Diversity generating retroelements are a genetic system used by bacteriophages (as well as bacteria and archea) to promote targeted hypervariability in genes (*Doulatov et al., 2004*). While DGRs are complex and consist of many components, at their most basic they can be identified as elements consisting of a reverse transcriptase gene and a repeated nucleotide sequence of length <150 bp that is found in two separate locations of the genome (*Doulatov et al., 2004*; *Minot et al., 2012*; *Schillinger et al., 2012*), termed the template region and the variable region. The template region is transcribed, then reverse transcribed in an error-prone fashion. The resulting cDNA is then integrated into the variable region, introducing base substitutions. Targeted hypervariation impacts functions including broadened host cell tropism by mutagenizing a phage tail fiber gene (*Doulatov et al., 2004*).

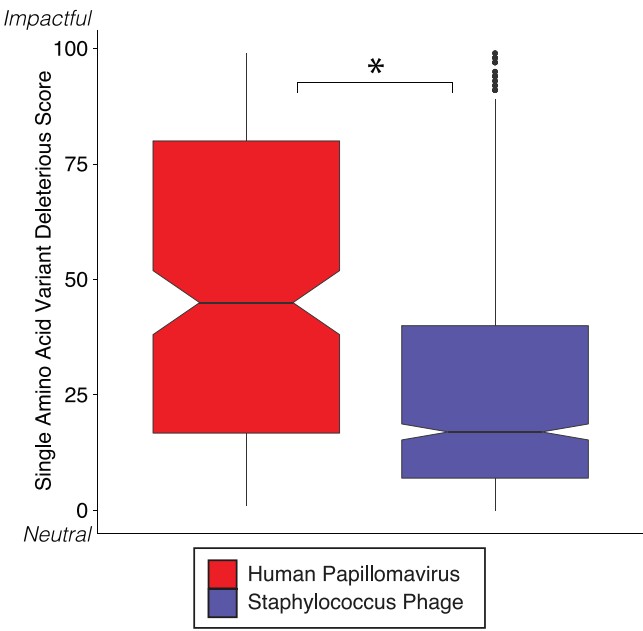

**Figure 3** **SVM predicted impact of hypervariable loci on phenotype.** Notched boxplot of deleterious scores in human papillomavirus (red) and *Staphylococcus* phage (blue) genomes. A low deleterious score indicates a predicted neutral phenotypic effect, while a high score indicates a predicted strong phenotypic effect. Asterisk indicates significant difference by Wilcoxon rank-sum test ($p < 1e15$). Boxplot parameters as described in Fig. 1.

We thus sought to identify candidate DGR cassettes within our viral contigs. We defined the candidate cassettes as pairs of non-overlapping regions with similar nucleotide sequences (tblastx of contigs against themselves, *e*-value $< 10e–50$) and co-localized on a contig containing a predicted virus/phage reverse transcriptase gene. We only considered cassettes that were located within a predicted viral gene, contained at least one hypervariable locus in their variable region, and exhibited truly random variation (different between reads). Based on these criteria, we identified one *Staphylococcus* phage DGR candidate that contained hypervariable loci. We also identified five other DGR candidates that were associated with hypervariable loci outside of predicted genes or that failed to demonstrate linkage disequilibrium, suggesting an association with cryptic genes or pseudogenes.

For the *Staphylococcus* phage DGR candidate with hypervariable loci, we calculated the linkage disequilibrium associated with the variable nucleotide positions to infer whether the DGR was active or inactive (e.g., an evolutionary artifact). The DGR cassette had unlinked nucleotide variation, which was supported by low levels of linkage disequilibrium (squared allelic correlation $R^2$) between SNP pairs in the variable region (Fig. 4). In this cassette, the template region has less frequent blocks of linkage equilibrium (unlinked variants) while the variable region was associated with greater linkage equilibrium. Together this suggests the observed *Staphylococcus* phage is active. The variable region was associated with a gene of unknown function.

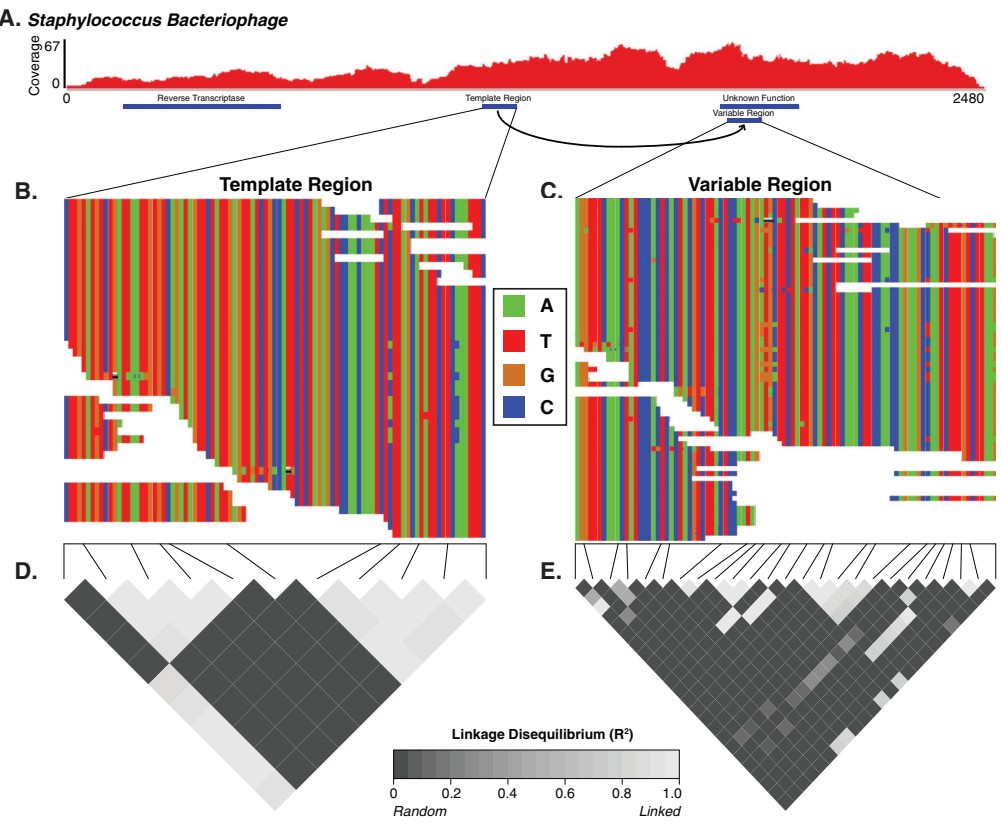

**Figure 4 The diversity generating retroelement as a mechanism for targeted nucleotide variation.** Alignment illustrating a putative diversity generating retroelement in *Staphylococcus* phage. (A) Sashimi plot of sequence coverage across the contig. Coverage ranges from 0 to $67\times$. Below the coverage is a map of the relevant genes predicted within the contig. Sequence alignment of the diversity generating retroelement template region (B) and variable region (C). Linkage disequilibrium heatmap for the template (D) and variable (E) region. Panels compare variable nucleotides to each other and darker tiles indicate decreased linkage disequilibrium correlation, according to squared allelic correlation ($R^2$) between pairs of SNPs.

## Skin virome variability patterns and SNP locations are reproducible across different datasets

We repeated our analyses in a separate, independently collected dataset from another research group (SRA BioProject: 46333) (*Oh et al., 2014*) to determine the generalizability of our findings. We analyzed metagenomic sequence data of skin specimens that were collected from the retroauricular crease without initial purification of virus like particles. Consistent with our primary analysis, *Staphylococcus* and *Propionibacterium* phages were identified as having the highest coverage and similarity to reference genomes (Fig. 5A). *Pseudomonas* phages were identified but were in the minority and had low coverage and similarity to reference genomes. HPV was not identified as a major virus in our analysis of the retroauricular crease; however, molluscum contagiosum virus, a poxvirus that causes cutaneous growths that become severe in immunocompromised states, was present in high relative abundance in agreement with the original published findings (*Oh et al., 2014*).

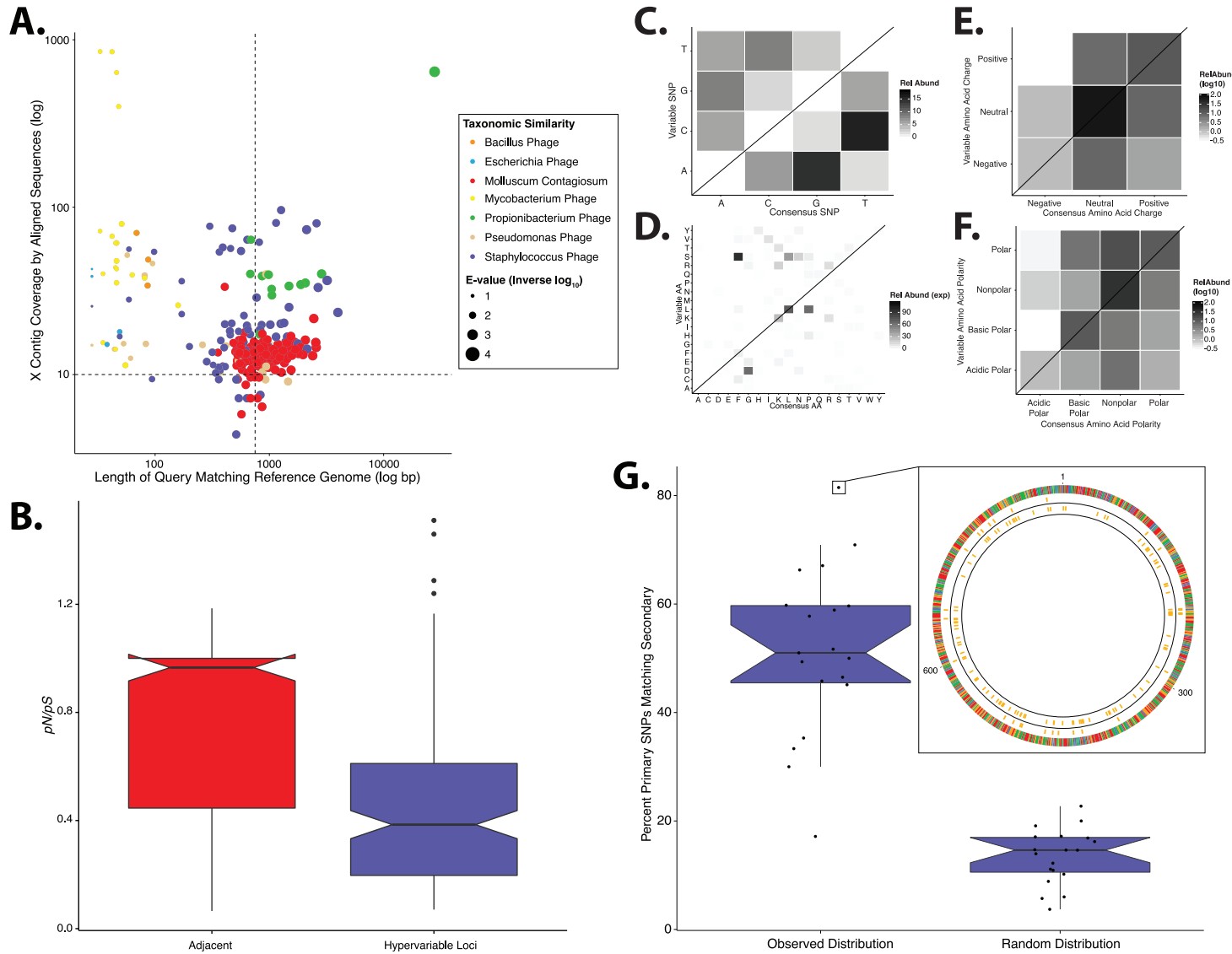

**Figure 5 Validation of study findings using secondary dataset.** Results from the Oh et al. (2014) dataset, which was analyzed using the same workflow as the primary dataset. (A) Scatter plot depicting the candidate contigs considered for analysis in this study. Each point is a contig that mapped to a reference virus genome. The *x*-axis shows the length (in nucleotides) of the contig subsection that mapped to the reference genome. The *y*-axis shows the overall coverage of the contig as a quantification of sequences aligning to the contig. The color highlights the reference virus genome that the contig was most similar to, and the size depicts the blast bit score associated with the contig-reference match. (B) Box plots depicting the evolutionary pressure of *Staphylococcus* bacteriophages at the hypervariable loci (blue) and the regions immediately adjacent to the hypervariable loci (red). (C) Heat map portraying the counts of every possible nucleotide substitution for each SNP found within 21 *Staphylococcus* phage hypervariable loci. Tile color weight corresponds to the relative abundance of SNP substitution counts. The diagonal line highlights the panels associated with no substitution. The substitution patterns of amino acids at each (D) SNP, (E) amino acid charge, and (F) polarity with acidity are also shown. (G) Notched boxplot illustrating the percent of primary dataset SNPs whose nucleotide positions were identical to those from the secondary validation sample set (left) compared to a simulated dataset of randomly assigned SNP locations (right). The inset shows an example contig identified in both datasets with 81% identical SNP positions. SNPs are represented as yellow lines, with the inner circle representing the validation dataset, and the middle circle representing the primary dataset. The outmost ring illustrates the contig, colored by nucleotides (A = red, C = blue, G = yellow, T = green). Boxplot parameters as described in Fig. 1.

Similar to our primary analysis, we identified 158 hypervariable loci within the *Staphylococcus* phage communities, and observed only 12 hypervariable loci associated with *Propionibacterium* phages, further highlighting the overall lack of genetic variability

of the *Propionibacterium* phage communities. The *Staphylococcus* phage hypervariable loci were associated with purifying selection, yielding a *pN/pS* ratio slightly below 0.4 (Fig. 5B), recapitulating our findings in the primary dataset.

*Staphylococcus* phage nucleotide substitutions were associated with transitions between guanine and adenine residues, as we observed in our primary analysis (Fig. 5C). The ti/tv ratio of these loci was 1.02. The most common amino acid substitution was proline to leucine (Fig. 5D) and the substitution properties appeared loosely specific based on charge and polarity (Figs. 5E and 5F), reproducing our findings (Fig. 2).

We also evaluated the reproducibility of SNP position between identical but independently assembled genomic contigs of the two studies. We quantified the proportion of SNPs in our primary dataset that were also found at the same position in the secondary dataset. This revealed a median of approximately 50% overlap between datasets (Fig. 5G). As a control, we generated a simulated dataset using randomly assigned SNP positions instead of those determined experimentally. This yielded a significantly lower median of approximately 15% shared nucleotide SNP calls (Fig. 5G), suggesting that the observed SNP position is not random. These data indicate that our findings are consistent across different skin virome populations and techniques of collection and sequencing.

## DISCUSSION

Here we report localized targeted hypervariability in some of the most prevalent members of the skin virome. Hypervariable loci provide a substrate for complex virus evolution throughout the virome, which manifest as natural selection that differs by virus type and enforces purifying selection. Hypervariable loci, which were present in genes encoding factors including virus tropism and host immune evasion, and were primarily under purifying selection, whereas overall virus genomes were under near neutral selection. We characterized selected substitution of nucleotides within hypervariable loci, with different variant patterns between HPVs and *Staphylococcus* bacteriophage communities. These findings were validated in an independently collected cohort.

We showed *Propionibacterium* phages exhibited strikingly low nucleotide variation with nearly no identifiable hypervariable loci. While this starkly contrasts with HPVs and *Staphylococcus* bacteriophages, it agrees with our current understanding of *Propionibacterium* phage diversity. Genome comparisons of *Propionibacterium* phage isolates revealed minimal nucleotide diversity, although this has yet to be supported by targeted metagenomic evidence such as presented here (*Marinelli et al., 2012*; *Liu et al., 2015*). The lack of *Propionibacterium* phage hypervariability in our metagenomic dataset provides another level of evidence for minimal *Propionibacterium* phage diversity on the skin.

There are several potential factors that could contribute to the limited diversity of *Propionibacterium* phages, and a consensus has yet to be reached. The lack of hypervariable loci suggests minimal evolutionary pressure on the phages, which may be a reflection of their environment. As suggested previously, the phages and their hosts reside in a unique and relatively isolated environment deeper in the skin, which may contribute to

low genomic diversity (*Marinelli et al., 2012*). Our data further support this hypothesis. Another factor that could contribute to differential phage genomic diversity is their host range. Although *Propionibacterium* phages have broad infectious capabilities within bacterial species, they may be limited in their ability to infect other species (*Marinelli et al., 2012*). *Staphylococcus* phages demonstrate greater genomic diversity, and may be capable of infecting a broader range of hosts.

We observed greater selective pressure on HPVs compared to *Staphylococcus* phages, which may reflect greater pressures from the human immune system, compared to phage bacterial hosts. This may also reflect the effects of different virus replication cycles on evolutionary selection. HPVs do not usually exist in a latent, integrated state, while *Staphylococcus* phages do (*Bae et al., 2006*; *Goerke et al., 2009*; *Edwards et al., 2013*). Our data suggest that at least one-half of the observed *Staphylococcus* phages have temperate replication cycles. As long as the *Staphylococcus* phages are integrated into the bacterial genome, we hypothesize that they are under less selective pressure by external factors.

The viral hypervariable loci are primarily associated with purifying selective pressure, a finding in agreement with previous non-metagenomic virus reports (*Chen et al., 2005*; *Wolf et al., 2006*; *Li et al., 2011*). The observed prominent purifying selective pressure supports an evolutionary model of long static periods punctuated by brief positive selection, as is observed in influenza virus (*Wolf et al., 2006*). Here nucleotide diversity acts as a primer for rapid virus adaptation through brief positive selection, while maintaining periods of consistency through purifying selection during static environmental conditions. As an example, some localized nucleotide diversity may allow for the generation of phages with different tropisms (e.g., different bacterial strains). If there are limited hosts, the phages that successfully infect those hosts will be selected for, and altered tropisms will be actively selected against. If that host population changes, then those viruses with the appropriate tropism will be selected for instead of being selected against. Ultimately, longitudinal and strain specific studies will be required to further address this hypothesis.

The amino acid substitutions associated with hypervariable loci were non-random and followed virus-specific substitution patterns (Fig. 2). HPV hypervariable loci were most associated with substitutions from glycine to valine. This substitution has recently been associated with infectious functionality, whereby the introduction of this mutation resulted in impaired infective ability of the virus (*Bronnimann et al., 2013*). This impaired infectious activity was attributed to a reduced efficacy of genomic DNA endosomal translocation within the host, which may have been the result of impaired trans-membrane alpha-helical self-association of the L2 minor capsid protein. Given these findings, our results suggest hypervariable loci are involved in promoting diversity in endosomal translocation motifs to some degree. Hypervariable loci may certainly have other diverse, functional roles, as evidenced by the wide range of hypervariable loci-containing genes.

The dominant amino acid substitution observed in *Staphylococcus* bacteriophages was from proline to leucine, a different substitution than that observed in HPV. This substitution could affect protein structure, particularly a loss of rigidity due to the loss of the proline ring structure. This observation may reflect a biologically important

adaptation of the bacteriophage to its *Staphylococcus* host, which have been shown to be auxotrophic for proline and leucine and may switch between auxotroph and prototroph depending on nutrient availability (*Emmett & Kloos, 1975*; *Nuxoll et al., 2012*). Because the amino acids may be in variable supply depending on the host, phages may alter their amino acid usage to exploit what is most readily available.

The overall selective nucleotide substitutions associated with HPV amino acid charge highlights a potential maintenance of HPV tropism. The lack of HPV substitutions between charges may suggest a selection against strong alterations in protein isoelectric points, which have been implicated in affecting HPV tropism (*Mistry, Wibom & Evander, 2008*). Furthermore, because acidic residues almost never mutated to non-polar residues, these acidic amino acids are potentially important external amino acids that may participate in tropic protein–protein interactions.

The described patterns in our findings suggest a role for targeted and/or localized genomic variation. One mechanism of such active targeted variation in *Staphylococcus* bacteriophages is DGRs. In this system, a phage-encoded reverse transcriptase copies a template region to create a variable region in a gene in an error-prone fashion. We identified such an element that is likely active and promotes diversity in a gene of unknown function. We additionally identified five other DGR elements whose hypervariable loci were not associated with an identified gene, suggesting an interesting phenomenon where high variability is selected for in non-coding regions. While informative, these discoveries only explain the diversity-generating mechanism of a small proportion of hypervariable loci. We suspect another underlying mechanism for the origin and evolution of other hypervariable loci could be that they are located on functionally important loci such as encoding regions that interact with other genes or are important to protein structure, therefore being functionally selected. Significant further investigation will be needed to characterize these and other potential mechanisms behind the observed hypervariable loci.

This study illustrates the diversity of evolutionary pressures on skin virus communities. It begins to provide further community-wide context to the molecular understanding of skin viruses, and highlights important aspects of their infectious cycles. These insights also contribute to understanding virus ecology of the human skin, and will inform future translational research into HPV vaccination, vaccination against other skin-associated viruses, effects of phages on bacterial pathogenesis, and phage therapy. Understanding how viruses evolve in their natural communities is crucial for improving these translational applications, and our findings here, which focus on HPV and *Staphylococcus* phages, will benefit cutaneous clinical virology and provide a foundation for future studies.

## CONCLUSION

We report that the skin virus communities contain hypervariable loci that are associated with strong purifying selection and targeted nucleotide substitution. The degree of selective pressure and impact of amino acid substitutions on protein chemistry (structure, isoelectric point, polarity) is virus specific, despite being members of the same community. These hypervariable loci are found within diverse viral strains, with varying

degrees of phylogenetic divergence over their evolutionary history. We further reproduce these findings in independently collected skin virus communities.

## ACKNOWLEDGEMENTS

We thank the members of the Grice and Bushman laboratories for their underlying contributions.

### Funding

This work was supported by grants from the NIH (NIAMS R00AR060873 to Elizabeth A. Grice and NIAMS R01AR066663 to Elizabeth A. Grice). Geoffrey D. Hannigan is supported by the Department of Defense, National Defense Science and Engineering Graduate fellowship program and Jacquelyn S. Meisel is supported by NIH T32 HG000046 Computational Genomics Training Grant. The funders had no role in study design, data collection and analysis, decision to publish, or preparation of the manuscript.

### Grant Disclosures

The following grant information was disclosed by the authors:
NIH: NIAMS R00AR060873 and NIAMS R01AR066663.
NIH Computational Genomics Training Grant: T32 HG000046.

### Competing Interests

Samuel S. Minot is an employee of One Codex.

### Author Contributions

- Geoffrey D. Hannigan conceived and designed the experiments, analyzed the data, wrote the paper, prepared figures and/or tables, reviewed drafts of the paper.
- Qi Zheng analyzed the data, reviewed drafts of the paper.
- Jacquelyn S. Meisel analyzed the data, reviewed drafts of the paper.
- Samuel S. Minot analyzed the data, reviewed drafts of the paper.
- Frederick D. Bushman analyzed the data, reviewed drafts of the paper.
- Elizabeth A. Grice conceived and designed the experiments, wrote the paper, prepared figures and/or tables, reviewed drafts of the paper.

### Data Deposition

   GitHub, ViromeVarScripts, https://github.com/Microbiology/ViromeVarScripts.

### Supplemental Information

Supplemental information for this article can be found online at http://dx.doi.org/10.7717/peerj.2959#supplemental-information.

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
