# Peer review of "Evolutionary and functional implications of hypervariable loci within the skin virome"

_PeerJ, doi:10.7717/peerj.2959_

## Round 0.1 · original submission · Minor Revisions

Please read and respond to comments from both reviewers in a point-by-point rebuttal, and resubmit your manuscript with the associated changes.

Also, please ensure that the English language in this submission meets our standards: uses clear and unambiguous text, is grammatically correct, and conforms to professional standards of courtesy and expression.

Reviewer 1 ·

Basic reporting

Appropriate

Experimental design

Appropriate

Validity of the findings

Needs work.

Additional comments

Hannigan et al. present their work describing hypervariable loci present in some viral members of the human skin microbiome. They utilize a new dataset and compare with older datasets to demonstrate that the same trends can be observed in each dataset. They identify human papillomaviruses, propionibacteria viruses, and staphylococcal viruses in their datasets and demonstrate that HPVs and staph viruses have hypervariable loci under purifying selection. While the data seem solid, I'm not sure the interpretation in each case is the most appropriate. It is a bit odd to be comparing small eukaryotic DNA viruses to large bacterial viruses, but those are the most interesting aspects of the authors' data. In the case of HPV, we know it's host, but I'm not sure we really know the host of the Staphylococcal viruses. There are many different species of staphylococci on the skin, and I wonder how much of the observed hypervariable loci represent viruses of different species or generalist viruses that can attack more than 1 staph species. My other specific comments are as follows:

Major
1. In discussion of the lack of diversity of the priopionibacteria phage, the authors leave out rather obvious conclusions, such as: Propionibacteria phages are likely limited to a single host species, while the staphylococcal phages may predate on many different species. There are numerous different staphylococci that inhabit the skin, including S. aureus, S. epidermidis, S. capitis, S. hominis, and many, many others. Are these phages generalists that are capable of infecting multiple different species? Are these phages specialists with limited host ranges, and thus, the authors are looking at many separate phages that have evolved to infect different staphylococcal species?
2. What type of staphylococcal viruses are we observing? Primarily lytic or lysogenic? This information may help to understand some of the patterns observed amongst the staphylococci and their viruses.

Minor
1. May benefit from adding in the metrics for the assemblies. The authors exclude a lot of data because of the <10X coverage. Is there a more appropriate middle ground that will allow the authors to characterize a larger proportion of the viruses present?
2. The authors are characterizing a highly uneven population given the relative numbers of viruses with <10X coverage compared to the 76,000 total contigs. Generally, this would be thought to be a side effect of the use of MDA, but MDA was not used. Authors should provide an explanation for the unevenness.
3. What is an adjacent region when calculating pN/pS?

·

Basic reporting

The basic structure of the manuscript is good and well written. In the form I only have three comments:
1) There is no legend or explanation for the Supplementary tables and figures. The tables has the header but from it is not clear what each column represents.
2) Due to the formatting of the main figure legends several spaces were deleted causing words to merge.
3) In figure 4 panel B, each block in the heat map correspond to the linkage equilibrium of how many bases? Is not a per nucleotide plot, but is not clear what a block size is and how it was determined.

Experimental design

The experimental design is adequate, regarding the methods, and in order to guarantee reproducibility some minor comments are:
1) It is not clear how a Hypervariable loci is identified. In the method section it says it uses a geometric distribution, and in the results section it is stated that: "Regions within each contig that contained a significantly higher frequency of SNPs over the stochastic background were identified as viral hypervariable loci." However, it is not clear how "significant" is defined, or how the "stochastic background" is defined.

2) The use of pN/pS is an interesting approach, in particular with the normalizations, the original form was published before, and although the modifications makes sense theoretically, would it be worth to validate the formula on a different/known dataset? To make sure is not creating any sort of bias.

3) Although the selective pressure is defined as pN/pS lower than 1, how is the significance of such deviation from 1 measured? Is not clear in the manuscript.

Validity of the findings

Few additional comments about the findings:

1) Figure 1: The 750bp cutoff seems relatively arbitrary, in particular since the figure shows that only few more bases to the left ~500bp (I guess) there is a more natural cutoff where the number of contigs and e-values drop. What is the justification for the 750bp cutoff?

2) Given the thresholds for contig size used for the phylogenetic analysis, it will be useful to know the size of the full-length protein coding region for L1 in HPV, and terminase large subunit for Staphylococcus and Propionibacterium. I assume it is essentially the same as the threshold used, otherwise it will be important to justify the threshold picked.

3) In search for DGR, among the different criteria, there is an inclusion parameter that is to only variable regions fall within coding genes. Is true that it has been described the usefulness of DGR for coding proteins, but if there are any such structure in what is not a predicted coding gene, it could be interpreted as either a cryptic or pseudogene not identified previously or some other very interesting phenomena where high variability is selected for non-coding genes. Given that only one DGR was found, I believe it could be interesting to mention at least if such sequences were found out of coding sequences.

4) Phrase in line 374 starting with “Hypervariable loci” suggest that the only hypervariable loci that were under purifying selection are those related to virus tropism and host immune evasion, and that is not completely clear from the results. First, given that I wasn't able to fully understand the supplementary tables, is not clear to me if all the hypervariable loci were identified in genes related to virus tropism or host immune evasion. Second, the purifying selection I believed is an average of all measurements observed, but is it true that all Hypervariable loci are under puryfing selection? It seems that the sentence is pushing a little the connection among those.

5) Is interesting that there are so many hypervariable loci but only 1 DGR, what could be the potential explanation for the origin and evolution of all the other hypervariable loci?

6) There is a very interesting observation in the results, where many hypervariable loci are found, and seem to be under purifying selection but the adjacent part of the genome is not (kind of counterintuitive), but then, the aminoacid substitutions observed in those regions tend to be mostly deleterious. What the explanation for this finding could be? Is said that the findings fit a model of long static periods. But given that the results are generated from pooled of human individuals, the specific viruses from each isolate should not necessarily be under the same evolutionary process, and why this type of selection is not observed in the rest of the viral genome? Why to keep hypervariable regions under purifying selection? It is still not clear to me.

---

## Round 0.2 · Minor Revisions

The revisions are very minor, but I would like to make sure that two reviewer suggestions are addressed:

1) I think it is important to include a figure describing what fraction of the total virome data in the study is being included in the more focussed analysis, as suggested by reviewer 2. A contig length vs coverage plot where the contigs that were included in later parts of the analysis are colored in a distinct way would accomplish this

2) Reviewer 1 makes a good point regarding the best way to identify whether a phage is lytic or lysogenic, and that blast homology to lysogens in the ACLAME prophage database may not accomplish this. In a 2011 paper from Rick Bushman's group (Minot et al Genome Research 21:1616), integrase content, homology with bacteria and homology with ACLAME database prophage proteins were all used. You might also consider running PHACTS or another machine learning algorithm to predict whether the phage protein sequences are likely to arise from lytic or lysogenic phages (https://www.ncbi.nlm.nih.gov/pmc/articles/PMC3289917/). However, the current approach is a reasonable start, please include a sentence about the limitations of BLAST homology to prophages as Reviewer 1 suggests.

Reviewer 1 ·

Basic reporting

Appropriate

Experimental design

Appropriate

Validity of the findings

Appropriate

Additional comments

May choose to include a sentence that classifying phage lifestyles based on blast analysis may result in errors if the genes in that database also are homologous to genes present in phages of differing lifestyles.

·

Basic reporting

No Comments

Experimental design

No Comments

Validity of the findings

No Comments

Additional comments

I think the manuscript has improved and all major and minor concerns were adequately addressed. With the final changes I have only one further comment:

1) Given that only the contigs with significant similarity to known reference genomes were used for the analysis in the current manuscript, and that the "dark matter" of the biology lies in all those viruses that have no known similarity to reference genomes. I believe is important to show to the reader what percentage of the total assembled contigs were the ones that had any similarity to known sequences. Ideally, maybe a plot like the ones the authors have done in previous studies of contig length vs contig coverage, where the contigs with significant similarity to reference genomes can be highlighted with a different color, could give the reader a general idea of how much of the viral diversity found in the samples were analyzed.

---

## Round 0.3 · accepted · Accept

Thanks for your careful attention to the reviewer comments, it is a well-conducted and worthwhile study that I look forward to seeing out in the world.

A few tiny and optional comments:

- the colon on line 122 has an awkward phrase starting with a capitalized letter before the three things being listed, please rephrase
- line 128 Staphylococcus not italicized
- line 237: what is the portion of contigs? ~500 out of nearly 250k?
- Supplemental Figure 1 in the filename label is about the pN/pS ratio and the new contig coverage figure does not have the figure number in the filename